# Do children allocated to different methods of complementary feeding introduction have distinct food preferences and flavor acceptance in the first years of life? A randomized clinical trial

**Renata Oliveira Neves**[1,2]*, **Elma Izze da Silva Magalhães**[2], **Cátia Regina Ficagna**[3]*, **Paula Ruffoni Moreira**[3], **Christy Hannah Sanini Belin**[3], **Larissa de Oliveira Silveira**[4], **Rogério Boff Borges**[5,6], **Leandro Meirelles Nunes**[3,7], **Juliana Rombaldi Bernardi**[1,3,7,8]

**1** Nutrition Department, Universidade Federal do Rio Grande do Sul, Porto Alegre, Brazil, **2** Department of Collective Health, Universidade Federal de Santa Maria, Santa Maria, Brazil, **3** Graduate Program in Child and Adolescent Health, Universidade Federal do Rio Grande do Sul, Porto Alegre, Brazil, **4** Medical School, Universidade Luterana do Brasil, Canoas, Brazil, **5** Unity of Biostatistics, Research Division, Hospital de Clínicas de Porto Alegre, Porto Alegre, Brazil, **6** Department of Statistics, Institute of Mathematics and Statistics, Universidade Federal do Rio Grande do Sul, Porto Alegre, Brazil, **7** Hospital de Clínicas de Porto Alegre, Porto Alegre, Brazil, **8** Graduate Program in Food, Nutrition, and Health, Universidade Federal do Rio Grande do Sul, Porto Alegre, Brazil

* ficagnacatia@gmail.com

## Abstract

This study aimed to analyze the food preferences and flavor acceptance among children allocated to different methods of complementary feeding (CF) introduction in the first years of life. This randomized clinical trial (RCT) involved three distinct groups of children regarding the method of CF introduction: Parent-Led Weaning (PLW); Baby-Led Introduction to SolidS (BLISS); and Mixed (both PLW and BLISS methods). The intervention occurred at 5.5 months old, and food preferences were analyzed between the ages of 12–35 months using the Food Preferences Questionnaire and the Taste Acceptance Test. The analysis was performed by intention-to-treat, using Pearson's chi-square test, Mann-Whitney test and Poisson regression. A d irected a cyclic g raph (DAG) was used to define the covariates. A total of 140 mother-infant pairs were randomized for the study (PLW n = 45; BLISS: n = 48; and Mixed: n = 47). Of them, 132 completed the Food Preferences Questionnaire, and 92 attended the Taste Acceptance Test. In unadjusted analysis, the prevalence of preferences for foods with a predominant sour taste was higher in the Mixed method compared to the PLW [Crude prevalence ratio (PR): 1.23; 95% CI: 1.03–1.47; p = 0.020]. However, after adjusting for covariates, this association did not remain statistically significant (Adjusted PR: 1.15; 95% CI: 0.94–1.41; p = 0.173). There was a significant association between the consumption of the solutions and their respective hedonic reactions in most of the offered tastes (sweet: p < 0.001; sour: p = 0.029; salty: p = 0.005;

**Data availability statement:** All relevant data are within the paper and Supporting Information files.

**Funding:** This research was supported by the Conselho Nacional de Desenvolvimento Científico e Tecnológico (CNPq), Brazil [grant number 407426/2021-3]; Fundo de Incentivo à Pesquisa e Eventos (FIPE), Hospital de Clínicas de Porto Alegre, Brazil [grant number 2019-0540]; Coordenação de Aperfeiçoamento de Pessoal de Nível Superior (CAPES), Brazil. The funders had no role in study design, data collection and analysis, decision to publish, or preparation of the manuscript.

**Competing interests:** The authors have declared that no competing interests exist.

umami: p = 0.026). In addition, food preferences related to the bitter taste were associated with the higher acceptance of the solution with the same taste in unadjusted analysis (Crude PR: 1.12; 95% CI: 1.1–1.25, p = 0.046), but this association did not remain significant in the adjusted analysis (Adjusted PR: 1.16; 95% CI: 0.99–1.37; p = 0.069). In conclusion, infants in the Mixed group showed higher sour taste preference than PLW, though not significant after adjustment.

## Introduction

Food preferences are shaped by the interaction of several intrinsic factors, such as birth weight, sex, and genetic predisposition; and environmental or family factors, such as income, ethnicity, socioeconomic level, and exposure to media [1,2], among others, as well as the early sensory experiences. The sensorial experiences of taste begin inside the uterus, in which the ingested amniotic fluid is flavored by the mother's food consumption [2].

Along with how long children were breastfed and whether they were eating foods of various flavors at an early age, the sensation of taste has become of great interest in recent years, as the major determinant of food acceptance patterns among children [3]. The acceptance of the tastes during the complementary feeding (CF) introduction can vary significantly between infants, being able to strongly influence food preferences [4,5].

Throughout the years, weaning strategies have been proposed to avoid unfavorable outcomes in children, such as food selectivity, from the traditional CF scheme, in which caregivers have greater interference, to methods with greater autonomy for the child, like the baby-led approaches [6]. In addition to responsive feeding, the methods of CF introduction have the potential to not only ensure a diet of nutritional adequacy but also promote optimal food-related behaviors and skills [7,8], as well as being related to the development of food preferences [6].

Previous research has suggested that infant-led weaning may be associated with a pattern of greater flavor diversity and that this would have a positive impact on the food preferences of infant-led weaning children. Therefore, this study aims to analyze the food preferences and flavor acceptance among children allocated to different methods of CF introduction in the first years of life.

## Methods

### Study design

A randomized clinical trial (RCT) entitled "Methods of Food Introduction in Children: a Randomized Clinical Trial", that involved three distinct groups of children regarding the method of CF introduction: (A) CF through conventional technique/Parent-Led Weaning (PLW); (B) CF using the Baby-Led Introduction to SolidS (BLISS) technique; and (C) CF using Mixed technique: both PLW and BLISS techniques. The study protocol and other relevant information were published elsewhere [9–17].

## Participants

Participants were recruited for the study via the Internet through social networks, websites, groups aimed at mothers, newspaper advertisements, and posters posted in appropriate places. The initial recruitment period occurred between March 2019 and October 2020.

Families residing in Porto Alegre and the metropolitan Brazilian region, with healthy, full-term, singleton newborns with a birth weight equal to or greater than 2500 grams, who had not yet started the introduction of foods, aged between 0–4 months of life, were considered eligible to participate in the study. For this outcome, children with food allergies or intolerances were excluded.

## Data collection

When entering the research (at 5.5 months of the child's life), mothers answered questions about sociodemographic, nutritional, and childcare-related characteristics. At 12 months, the Food Preferences Questionnaire was applied using an online form. Between 12 and 35 months of age, a face-to-face meeting was held for the application of the taste acceptance test.

## Intervention

According to the method of CF introduction to which they were randomized, the intervention was carried out when the child was 5.5 months old, in groups of 4–8 families at a time, with an average duration of three hours. Twenty-four intervention sessions were carried out to train all families. The interventions on each method of CF introduction were carried out on different days, and each participant received the intervention once, which consisted of a practical workshop in a nutrition clinic with an experimental kitchen, where a nutritionist explained how the introduction of foods should occur. Food preparations with examples of meals were performed. Families discovered which method of CF introduction they were allocated to only at the time of the intervention.

All groups received guidance on maintaining exclusive breastfeeding practice until the child was six months old, continuing breastfeeding after until two years of life or more, and the importance of a healthy diet with natural and minimally processed foods, in addition to guidance on not offering sugar and ultra-processed foods before the age of two.

The specific information that each group received was:

- PLW: The families were encouraged to feed their infants with food in a puree form, with the help of a spoon. Families were instructed to gradually progress the food consistency until it reached the texture usually offered within the family, at around 12 months of age. Families were also instructed not to mix foods, to ensure that the infant was able to learn the difference between flavors.

- BLISS: Caregivers assigned to this method learned to prepare foods in the form of strips or sticks, allowing the child to self-feed without adult interference. Despite this, the importance of adult supervision at mealtimes was emphasized.

- Mixed: This group was instructed to combine the two methods explored above, according to the wishes of the child. The family was instructed to initially offer foods using the BLISS technique. If the child showed dissatisfaction or disinterest in food, according to the BLISS technique, they were instructed to offer the food using the PLW technique at the same meal.

## Outcomes

The RCT had as primary outcomes the z-scores of anthropometric measurements (weight for age, weight for length, length for age, and BMI for age z-scores) [9,10] to investigate child growth and nutritional status [18]. The secondary

outcomes, defined a priori in the RCT, included, among others (prevalence of choking, dietary variety, child and parental eating behavior, iron deficiency, oral hygiene behavior, dental caries, dental development, gingival health, prevalence of functional constipation, maternal perception of CF methods, and prevalence of childhood eating disorders), food and flavor preferences [9,10], which were investigated as outcomes in the present study.

## Food preferences questionnaire

The Food Preferences Questionnaire was translated from Schwartz et al. [19], a previously validated instrument for assessing children's food acceptance, with examples of food groups according to their predominant taste (sweet, sour, salty, umami, and bitter), and was analyzed according to a 5-point scale varying between strong rejection and strong acceptance. An example of a food that was not commonly consumed in Brazil was removed from the test at the time of translation. The food groups were considered as follows, according to Schwartz et al. [19]:

- Predominantly sweet flavor: unsalted mixed vegetables, vegetables with cereal; carrots, cabbage, beans; pear, apple, banana; biscuits, porridge, infant's cereal; natural juice, flavored yogurt, and dairy drink.

- Predominantly sour flavor: tomato, eggplant; lemon, natural yogurt, pineapple; apricot, orange, peach, strawberry.

- Predominantly salty flavor: salted vegetables, vegetables with broth, stew; spinach, turnips, broccoli, salted chicory; ham, French fries, cheese, bread, snacks.

- Predominantly umami flavor: vegetables with broth and stew.

- Predominantly bitter flavor: carrots, cabbage, beans; spinach, turnips, broccoli, salted chicory.

For the analysis, the following categorization for the 5-point scale was adopted: rejection (points 1–3) and acceptance (points 4 and 5).

## Taste acceptance test

The taste acceptance test was applied face-to-face, between 12 and 35 months of age, by a blinded researcher regarding the method of CF introduction to which the child was allocated. The Taste Acceptance Test was translated from Schwartz et al. [20] with the offer of solutions for each of the basic tastes. The solutions were expected to generate a moderate taste and were composed of 50mL of mineral water and the flavoring agents: 3.42g of lactose for the sweet taste, 0.057g of citric acid for a sour taste, 0.25g of sodium chloride for a salty taste, and 0.085g of monosodium glutamate for an umami taste; and 15mL of mineral water with 0.16g of urea for a bitter taste; and were offered in this order. To limit the variability of thirst and hunger and their potential impact on the acceptance of different tastes, parents were instructed to offer milk or water 30 minutes before the test, and then not to offer any food or drink.

Each solution was offered for 30 seconds, with a 15-second pause, and pure water was offered, followed by a 15-second pause for the next solution. They were offered by a blinded researcher and measured before and after the test. The evaluation of the test was carried out in two ways: calculation of the ingested volume and reaction of the infant by the hedonic scale. This was based on the infant's facial and body reactions, on a 5-point scale: 1) strong rejection (child cried at first sip), 2) mild rejection (child, at first sip, spitting, frowning, or pulling away the container and stop drinking), 3) neutral reaction (child accepted the drink two times or two sips, and after frowning and stops drinking), 4) mild acceptance (drinking more than two sips, without specific reaction), and 5) strong acceptance (accepting the drink at first and demonstrating well-being when ingesting the drink, such as a relaxed face, smiling or pulling the glass towards her) [20]. For a better perception of the reactions, the test was filmed and later analyzed in duplicate.

For the analysis, the 5-point scale was categorized as rejection (points 1, 2and3) and acceptance (points 4 and 5).

                                                                                    

## Sample size

The sample size calculation was performed for the primary outcome of the RCT, according to the previously mentioned publication, contemplating studies already published on the subject, according to the main aim of the clinical trial [21,22]. Considering a unit standard deviation equal to 1.4, with a power of 80% and a significance level of 5%, the sample calculation resulted in 48 mother-infant pairs for each of the three intervention groups, totaling 144 pairs of mothers and their respective children.

## Randomization

The randomization of the participants took place in blocks of three arms, until reaching the calculated number of mother-infant pairs, generated by the computer on the website www.randomization.com. The families became aware of the group to which they were allocated only on the day of the intervention.

## Statistical analysis

All analyses were performed by intention-to-treat. The database was created utilizing the SPSS® Program, Statistical Package for the Social Sciences version 29.0, with double entry and subsequent validation.

Categorical variables were described using absolute and relative frequencies [(n (%)] and numeric variables were described as measures of central tendency and dispersion (mean and standard deviation; median and P25- P75). The distribution of numeric variables was assessed through visual inspection of histograms, verification of asymmetry and kurtosis measures, as well as the application of the Shapiro-Wilk test. To compare the sociodemographic, nutritional, and childcare-related characteristics of the study sample between the intervention groups (PLW, BLISS and Mixed), Pearson's chi-square test was used for the proportions of categorical variables, and the Mann-Whitney test was used for the medians of numerical variables. The Mann-Whitney test was also used to compare the medians of the volume consumed in the taste acceptance test according to the infant's reaction (Acceptance and rejection) to the same substances. A significance level of 5% (p ≤ 0.05) was considered for all analyses.

Poisson regression analysis with robust variance was used to estimate prevalence ratios and their 95% confidence intervals for the associations between the method of CF introduction and food preferences according to the predominant taste, as well as between food preferences according to the predominant taste and reactions in the taste acceptance test. To control confounding, Poisson regression analysis was adjusted for potential confounders. A minimum adjustment set of variables to control for confounding in each association evaluated was identified based on a directed acyclic graph (DAG) constructed using the online DAGitty® 3.2 software, applying the backdoor criterion [23]. The DAG for estimating the total effect of the method of CF introduction on food preferences according to the predominant taste indicated the need for adjustment for m aternal education, parity, maternal BMI, daycare attendance and child's primary caregiver (S1 Fig).

The DAG for estimating the total effect of the food preferences according to the predominant taste on reactions in the taste acceptance test, in turn, indicated the need for adjustment for: Maternal education, maternal BMI, age at CF introduction, method of CF introduction, age at the application of the food preferences questionnaire and taste acceptance, daycare attendance, child's primary caregiver and type of milk consumed (S2 Fig).

## Ethical aspects

The anonymity of the participants was ensured through the process of deidentification, where all personal identifiers were removed or replaced to protect their identities. Additionally, all participants provided their informed consent by signing the Free and Informed Consent Form. This study was submitted to and approved by the Research Ethics Committee of the Hospital de Clínicas de Porto Alegre (CAAE: 1537018500005327).

### Registration

After study approval by the Research Ethics Committee, the RCT was registered at https://ensaiosclinicos.gov.br/rg/RBR-229scm, identifier [RBR-229scm U1111-12269516]. The authors confirmed that all ongoing and related trials for this intervention are registered.

### Results

A total of 140 mother-infant pairs were randomized for the study, 45 (32.1%) in the PLW group, 48 (34.3%) in the BLISS group, and 47 (33.6%) in the Mixed group. Of them, 132 completed the Food Preferences Questionnaire and were included in the study, and 92 attended the Taste Acceptance Test. The study flowchart was presented in Fig 1.

The descriptive data of the sample are presented in Table 1. Of the 132 mothers, 112 (84.8%) considered themselves white ethnicity, 113 (85.6%) lived with their partner at the moment of the study, had a median of 18 [15,20] years of study, and a median monthly family income of R$6,000.00 [4,000.00; 10,000.00]. Moreover, 108 (81.8%) were primiparous and half (50.8%) of the infants were female. The median maternal age in the intervention was 33.5 [30.0; 36.0] years, and half (52.0%) of them had a BMI above the recommended.

Regarding the child characteristics, 180 days were the medians of exclusive breastfeeding and the moment of CF introduction (interquartile range [150; 180] and [180; 180], respectively). At twelve months, 93 (70.5%) still didn't attend daycare, the mother was the main caregiver of 118 (89.4%) of the children and 69 (52.3%) consumed breast milk as the only type of milk. No differences were detected between the CF groups in any of these variables (p > 0.05).

In the Food Preferences Questionnaire, 97% of participants accepted sweet foods, 87.8% accepted sour foods, 92.1% accepted salty foods, 82.9% accepted umami foods, and 90.2% accepted bitter ones. The crude and adjusted prevalence ratios (PR) of the association between the method of CF introduction and the food preferences according to the predominant taste are shown in Table 2. In adjusted analysis, the prevalence of preferences for foods with a predominant sour taste were 23% higher (Crude PR: 1.23; 95% CI 1.03–1.47; p = 0.020) in the Mixed method compared to the PLW, however, after adjusting for covariates, this association did not remain statistically significant (Adjusted PR: 1.15; 95% CI: 0.94–1.41, p = 0.173). The preference for foods with other predominant tastes was not associated with the method of CF introduction, neither in the unadjusted nor in the adjusted analysis.

When observing the Taste Acceptance Test, 40.7% (n = 33) of the participants rejected the sweet taste, 64.7% (n = 47) rejected the sour taste, 54.8% (n = 37) rejected the salty taste, 59.2% (n = 34) rejected the umami taste, and 66.7% (n = 33) rejected the bitter taste. The median consumption quantity of the solutions was 5.0mL, 4.0mL, 3.5mL, 4.0mL, and 5.0mL, respectively. There were no statistical differences between the CF groups (p > 0.05). There was an association between the consumption of the solutions and their respective hedonic reactions in most of the offered tastes (p < 0.001 in sweet taste, p = 0.029 in sour taste, p = 0.005 in salty taste, p = 0.026 in umami taste); however, there was no association in bitter taste (p = 0.811) (Table 3).

The crude and adjusted PR of the association between the food preferences according to the predominant taste and the reactions in the taste acceptance test are shown in Table 4. In unadjusted analysis, food preferences related to the bitter taste were associated with higher acceptance of the solution with the same taste (Crude PR: 1.12; 95% CI: 1.01–1.25, p = 0.046). However, in adjusted analysis, this association lost its significance (Adjusted PR: 1.16; 95% CI: 0.99–1.37; p = 0.069).

### Discussion

The PLW method of CF introduction presented a lower preference for sour-tasting foods compared with the Mixed method in the crude analysis. However, this association did not remain statistically significant after the adjusted analysis. No associations were observed for foods with other predominant tastes, either in the crude or adjusted analyses. The findings should therefore be interpreted with caution, as they suggest only a modest influence of CF methods on taste

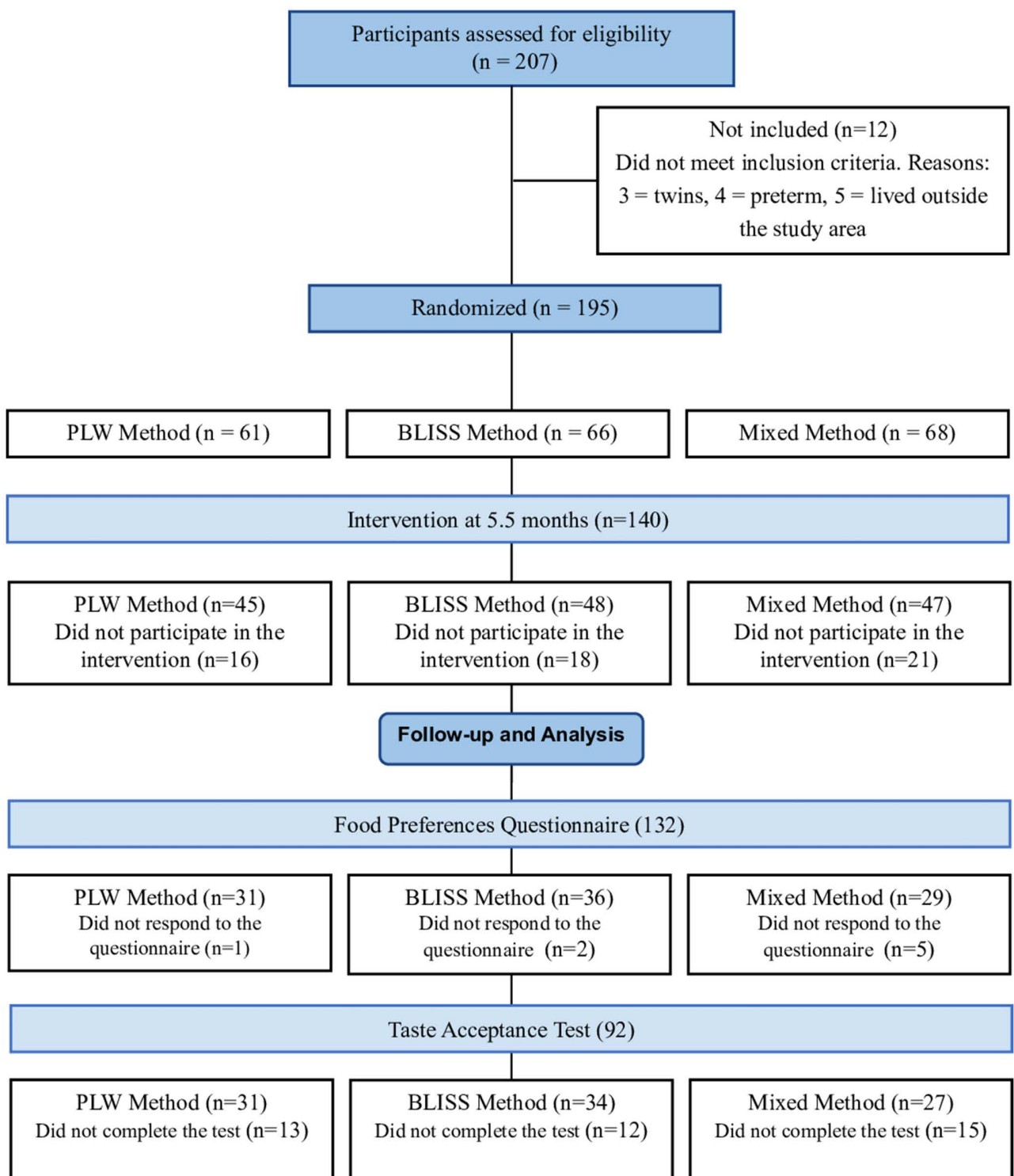

**Fig 1. Study flowchart of the randomized clinical trial.** Subtitle: PLW = Parent-Led Weaning; BLISS = Baby-Led Introduction to SolidS. Recruitment and randomization were made online. Intervention and the Food Preferences Questionnaire were administered in person or online. The Acceptance Test was made in person.

**Table 1. Sociodemographic, nutritional, and childcare-related characteristics of the study sample, according to the method of complementary feeding introduction. Porto Alegre, Brazil.**

| Variables | Total (n = 132) | PLW (n = 44) | BLISS (n = 46) | Mixed (n = 42) | p |
|---|---|---|---|---|---|
| *Baseline variables* | | | | | |
| **Maternal ethnicity** | | | | | |
| White | 112 (84.8) | 36 (81.8) | 39 (84.8) | 37 (88.1) | 0.719 |
| non-white | 20 (15.2) | 8 (18.2) | 7 (15.2) | 5 (11.9) | |
| **Maternal marital status** | | | | | |
| With partner | 113 (85.6) | 35 (79.5) | 42 (91.3) | 36 (85.7) | 0.283 |
| Without partner | 19 (14.4) | 9 (20.5) | 4 (8.7) | 6 (14.3) | |
| **Maternal education (years)** | 18.0 [15.0; 20.0] | 16.5 [13.0; 20.0] | 18.0 [15.0; 20.0] | 18.0 [16.0; 20.0] | 0.336 |
| **Family income (BRL)*** | 6,000.0 | 5,000.0 | 8,000.0 | 5,500.0 | 0.294 |
| | [4,000.0; 10,000.0] | [3,775.0; 10,000.0] | [4,000.0; 14,000.0] | [3,875.0; 10,000.0] | |
| Missing data (n) | 1 | 0 | 1 | 0 | |
| **Parity** | | | | | |
| Primiparous | 108 (81.8) | 34 (77.3) | 36 (78.3) | 38 (90.5) | 0.210 |
| Multiparous | 24 (18.2) | 10 (22.7) | 10 (21.7) | 4 (9.5) | |
| **Child's sex** | | | | | |
| Male | 65 (49.2) | 20 (45.5) | 22 (47.8) | 23 (54.8) | 0.670 |
| Female | 67 (50.8) | 24 (54.5) | 24 (52.2) | 19 (45.2) | |
| **Maternal age (years)** | 33.5 [30.0; 36.0] | 33.0 [27.0; 36.0] | 34.5 [31.0; 38.0] | 32.0 [28.0; 35.0] | 0.113 |
| **Maternal BMI*** | | | | | |
| Eutrophic | 49 (48.0) | 16 (45.7) | 20 (55.6) | 13 (41.9) | 0.508 |
| Overweight/Obesity | 53 (52.0) | 19 (54.3) | 16 (44.4) | 18 (58.1) | |
| Missing data (n) | 30 | 9 | 10 | 11 | |
| **EBF duration (days)** | 180.0 [150.0; 180.0] | 180.0 [153.0; 180.0] | 180.0 [90.0; 180.0] | 180.0 [152.0; 180.0] | 0.812 |
| **Age at CF introduction (days)** | 180.0 [180.0; 180.0] | 180.0 [171.0; 180.0] | 180.0 [180.0; 180.0] | 180.0 [180.0; 180.0] | 3 |
| *Infant variables at 12 months* | | | | | |
| **Daycare attendance** | | | | | |
| Yes | 39 (29.5) | 18 (40.9) | 12 (26.1) | 9 (21.4) | 0.115 |
| No | 93 (70.5) | 26 (59.1) | 34 (73.9) | 33 (78.6) | |
| **Child's primary caregiver** | | | | | |
| Mother | 118 (89.4) | 42 (95.5) | 40 (87.0) | 36 (85.7) | 0.270 |
| Other | 14 (10.6) | 2 (4.5) | 6 (13.0) | 6 (14.3) | |
| **Type of milk consumed** | | | | | |
| BF | 69 (52.3) | 19 (43.2) | 28 (60.9) | 22 (52.4) | 0.194 |
| Formula/cow milk | 31 (23.5) | 9 (20.5) | 10 (21.7) | 12 (28.6) | |
| BF + formula/cow milk | 32 (24.2) | 16 (36.4) | 8 (17.4) | 8 (19.0) | |

Legend: n = number of participants; PLW = Parent-Led Weaning; BLISS = Baby-Led Introduction to SolidS; BMI = body-mass-index; EBF = exclusive breastfeeding; CF = complementary feeding; BF = breastfeeding.

Statistical tests: Pearson's Chi-square, for qualitative variables (expressed with n (%)), and Mann Mann-Whitney test for quantitative variables (expressed with median [P25; P75]).

Some variables, highlighted with an asterisk (*), may not totalize 132 participants, due to missing data.

preferences. Moreover, an association between the Taste Acceptance Test and the Food Preferences Questionnaire was observed in the crude analysis for the bitter taste; nevertheless, these results may serve as an indication warranting further research with a larger sample on how CF methods could influence children's taste preferences.

**Table 2. Crude and adjusted prevalence ratios of the association between the method of CF introduction and the food preferences according to the predominant taste (n = 132).**

| Tastes | Crude PR (95% CI) | p | Adjusted PR (95% CI) | p |
|---|---|---|---|---|
| **Sweet** | | | | |
| PLW | 1 | | 1 | |
| BLISS | 1.07 (0.99–1.16) | 0.083 | 1.10 (1.00–1.21) | 0.058 |
| Mixed | 1.05 (1.00–1.15) | 0.326 | 1.06 (0.96–1.18) | 0.177 |
| **Sour** | | | | |
| PLW | 1 | | 1 | |
| BLISS | 1.18 (0.98–1.42) | 0.075 | 1.15 (0.94–1.41) | 0.167 |
| Mixed | 1.23 (1.03–1.47) | **0.020** | 1.15 (0.94–1.41) | 0.173 |
| **Salty** | | | | |
| PLW | 1 | | 1 | |
| BLISS | 0.90 (0.79–1.04) | 0.143 | 0.96 (0.84–1.09) | 0.540 |
| Mixed | 0.99 (0.90–1.10) | 0.941 | 1.00 (0.89–1.12) | 0.841 |
| **Umami** | | | | |
| PLW | 1 | | 1 | |
| BLISS | 0.86 (0.71–1.04) | 0.109 | 0.88 (0.73–1.06) | 0.188 |
| Mixed | 0.84 (0.68–1.03) | 0.087 | 0.80 (0.63–1.00) | 0.050 |
| **Bitter** | | | | |
| PLW | 1 | | 1 | |
| BLISS | 1.03 (0.90–1.18) | 0.674 | 1.06 (0.91–1.25) | 0.423 |
| Mixed | 1.02 (0.88–1.18) | 0.780 | 0.99 (0.81–1.21) | 0.905 |

Legend: PR = Prevalence Ratio; 95% CI = 95% Confidence Interval; PLW = Parent-Led Weaning; BLISS = Baby-Led Introduction to SolidS.

*= Adjusted for: Maternal education, parity, maternal BMI, daycare attendance and child's primary caregiver.

Statistical model: Poisson regression with robust variance.

**Table 3. Median (P25; P75) of the volume consumed in the Taste Acceptance Test, according to the infant's reaction to the same substances (n = 92).**

| Tastes | Acceptance | Rejection | p |
|---|---|---|---|
| **Sweet (n = 81)** | 7.0 [4.1; 9.9] | 3.5 [2.5; 6.0] | **<0.001** |
| **Sour (n = 73)** | 6.0 [3.3; 10.6] | 3.5 [2.5; 6.0] | **0.029** |
| **Salty (n = 68)** | 6.3 [3.0; 13.5] | 3.0 [2.0; 4.0] | **0.005** |
| **Umami (n = 57)** | 6.0 [4.0; 9.0] | 3.5 [2.3; 6.0] | **0.026** |
| **Bitter (n = 49)** | 5.0 [2.5; 7.0] | 5.0 [3.0; 7.5] | 0.811 |

Statistical test: Mann-Whitney (expressed with median [P25; P75]).

The preference for food groups according to their predominant taste was practically total in the sweet taste, and equally high in the other tastes. Analyzing separately according to the method of CF introduction, the Mixed group showed a 23% higher prevalence of preferring sour-tasting foods when compared to the PLW group. The existing literature demonstrates a lack of articles analyzing mixed methods of CF introduction, making it difficult to compare this result with those from other research.

Despite this, the literature already demonstrates that, in baby-guided methods, it can be easier to offer citrus fruits, which are difficult to mash and offer in puree form at the beginning of the CF introduction, as demonstrated by a longitudinal study carried out in Ireland, where infants who accepted sour solutions also had a higher fruit intake at six months,

**Table 4. Unadjusted and adjusted prevalence ratios of the Food Preferences Questionnaire, according to the Taste Acceptance Test reaction (n = 92).**

| Tastes Reaction | Unadjusted | | Adjusted * | |
|---|---|---|---|---|
| | Crude PR (95% CI) | p | Adjusted PR (95% CI) | p |
| Sweet | 1.01 (0.94–1.09) | 0.794 | 1.02 (0.93–1.12) | 0.653 |
| Sour | 1.06 (0.90–1.26) | 0.488 | 1.06 (0.85–1.31) | 0.613 |
| Salty | 1.11 (0.97–1.27) | 0.134 | 1.14 (0.99–1.31) | 0.060 |
| Umami | 0.92 (0.69–1.22) | 0.550 | 0.90 (0.69–1.17) | 0.426 |
| Bitter | 1.12 (1.01–1.25) | **0.046** | 1.16 (0.99–1.37) | 0.069 |

Legend: PR = Prevalence Ratio; 95% CI = 95% Confidence Interval.

*=Adjusted for: Maternal education, maternal BMI, age at CF introduction, method of CF introduction, age at the application of the food preferences questionnaire and taste acceptance, daycare attendance, child's primary caregiver and type of milk consumed.

Statistical model: Poisson regression with robust variance.

and a significantly higher increase in their fruit intake from 12–18 months [24]. Another study, a cohort comparing food preferences in infants following traditional spoon-feeding and BLISS, showed that the second method was exposed to more varied and textured foods from an early age, and an increased variety in fruit and vegetable intake was apparent by two years of life [25]. At the end of the second year, the taste intensity and variety of the diet increase and become more similar to that of adults [26].

Other factors that the literature shows lead to the preference of the flavors are exposure during the prenatal period, early milk-feeding, repeated food exposure, parental role modeling, and the type of milk consumed after one year of life [27–29]. A complementary investigation, seeking to relate the type of milk consumed and the tests performed, did not find significant differences in the current sample. An analysis from a prospective birth cohort with 10 years of follow-up study, in agreement with our results, did not detect a significant association between early life feeding practices and feeding patterns at school age. The authors implied that early feeding practices are intrinsically related to socioeconomic patterns, which can equally influence later feeding patterns [30].

The prevalence of negative reactions in the Taste Acceptance Test was smaller in the sweet taste and larger in the sour and bitter tastes, without association. This result is in line with the current literature, which evidence that children are born with an innate preference for sweet flavors, while they show refusal behaviors for sour and bitter tastes [2,31,32]. A longitudinal study that evaluated taste acceptance in the first two years of life reported that the acceptance of saltiness, sweetness, sourness, and umami tastes increased sharply over the first year of life. However, between 12 and 20 months, the acceptance of these tastes decreased, except for the sweetness; while the bitter taste acceptance remained constant [33].

Despite the author's unable to express differences between CF groups on the Taste Acceptance Test, research conducted in the United Kingdom indicated that infants who had their CF introduced by baby-led techniques were more exposed to vegetables and protein-rich foods, while those who introduced foods via traditional method were more exposed to complete meals, with different food groups at the same moment [34]. Nuzzi et al. [6] emphasized that the benefits of baby-led techniques depend on the family diet; therefore, if the family's diet is inadequate, this method can expose the child to the risk of excessive consumption of sodium, saturated fats, and proteins.

Also, on the Taste Acceptance Test, the consumed amount of the solution and the infant's hedonic reaction were associated with most flavors, except for the bitter taste. In reality, the child's hedonic reaction cannot always be faithful to its acceptance, since, from an evolutionary perspective, innate hedonic facial expressions play an important adaptive role, allowing infants to convey information about the sensory characteristics of food to the caregivers [35]; not necessarily meaning rejection of the same. That is, eating leads to physiological activations, and these produce hedonic experiences

[36]. Considering this, this inaccurate relationship between reactions and acceptance may have impacted the current study.

Nevertheless, while practically the entire sample responded to the online Food Preferences Questionnaire, 69.7% of the sample took the Taste Acceptance Test in person, and of those who performed the test, not all agreed to consume all the solutions, which may have potentially reduced the chance of the statistical test demonstrating associations with the Food Preferences Questionnaire. This variability, due to some children not having tried all the solutions, may have occurred due to the collection location and the glasses and bottles being from the research utensils, both of which were previously unknown to them. However, Indrayan & Mishra [37] emphasize that studies may provide more truthful results with small samples, because intensive efforts can be made to control all the confounders and how they operate, obtain more accurate data.

In the complementary analysis of this study, a crude association was observed between bitter taste preferences and higher acceptance of the same flavor in the Taste Acceptance Test. However, this association did not remain significant after adjustment. In addition to that, the OPALINE French birth cohort exposed in their results that, at weaning, a higher preference for sweet, sour, and umami tastes is associated with a higher acceptance of sweet-, sour-, and umami-tasting foods, respectively; and that both taste and flavor preferences evolve during the first years of life [38]. Regardless of this outcome, when adjusted for the sample characteristics exposed in the DAG, the association lost its significance. This lack of association may be due to the confounding effect of the included variables or the low sample power caused by losses.

Still on the association between taste and food preferences, children allocated to a large European study (the I. Family study) showed that a higher quality diet during the CF period was associated with lower chances for a high-sweet and a high-fat taste preference, and increased chances for a high-bitter taste preference. The authors suggest that current food choices can mold children's preferences for sweet, fatty, and bitter tastes, independent of their infant feeding methods [39]. Also, a systematic review showed that the effect of repeated exposure to healthy food can be generalized to other foods in the same category [40], this being the most effective way to promote the acceptance of distinct flavors and foods [41]. Other aspects that can impact the variation of taste acceptance are the variations in gustatory perception, with the influence of genetic polymorphisms [42–45], and the state of hunger, when food becomes more palatable and there is more sensitivity to sweet, sour and salty tastes [4].

This study indicates some limitations. The children were assessed at different ages, between 12 and 35 months, which occurred due to delays and difficulties imposed by the COVID-19 pandemic, leading to losses in follow-up after 12 months. These losses may have been related to the interruption of in-person data collection and families' concerns about exposing their children to non-mandatory health services. In addition, the sociodemographic profile of the sample was higher than that of the general Brazilian maternal population, as it was predominantly composed of mothers with higher educational levels and income. Therefore, the findings have limited external validity and cannot be generalized to the entire Brazilian population. The intention-to-treat analysis assessed the effect of the intervention on taste preferences and acceptance, but not the adherence to the method. The Food Preferences Questionnaire, as a self-reported questionnaire, may be subject to the caregiver's subjective interpretation of the child's acceptance or rejection of foods. For the Taste Acceptance Test, factors such as the testing environment, use of unfamiliar utensils, offer of artificial solutions instead of foods, and absence of usual social mealtime influences may affect reactions. Both instruments capture specific aspects of food acceptance and preference, but do not replace observation in natural settings. There were participant losses in the food preferences questionnaire and taste acceptance test. However, we performed an analysis comparing the baseline characteristics between completers and non-completers of the food preferences questionnaire and the taste acceptance test and no statistically significant differences were observed between the groups (S1 Table). Therefore, attrition bias is unlikely to influence the results. On the other hand, given the multiple comparisons carried out, there is an increased risk of Type I error. This is particularly relevant as significant findings were limited to sour taste preferences, which should be interpreted with caution.

Despite these limitations, this study, to the authors' knowledge, is the first RCT to randomize and follow participants in three different methods of CF introduction, providing important insights into how methods of CF introduction can influence taste preferences and acceptance in children in early childhood.

## Conclusions

Infants allocated to the Mixed group showed a greater preference for sour taste compared with the PLW group. However, this association did not remain statistically significant after the adjusted analysis and should therefore be interpreted with caution. The association observed between bitter taste preferences and the acceptance of solutions with the same taste also did not remain after adjustment, reinforcing the preliminary nature of these findings. Future longitudinal research is needed to better explore infant feeding behaviors in relation to adherence to CF methods and to include foods specific to the Brazilian cultural context.

## Supporting information

**S1 Fig. Directed Acyclic Graph for estimating the total effect of the method of CF introduction on food preferences according to the predominant taste.** Legend: CF = complementary feeding; BMI = Body Mass Index; EBF = exclusive breastfeeding.
(TIFF)

**S2 Fig. Directed Acyclic Graph for estimating the total effect of the food preferences according to the predominant taste on reactions in the taste acceptance test.** Legend: CF = complementary feeding; BMI = Body Mass Index; EBF = exclusive breastfeeding.
(TIFF)

**S1 Table. Comparison of the characteristics baseline of completers and non-completers of the food preference questionnaire and the taste acceptance test in the study.**
(DOCX)

**S1 Dataset. Minimal data set of the study.**
(XLSX)

**S1 Checklist. CONSORT 2025 editable checklist.**
(DOCX)

## Acknowledgments

The authors thank the researchers for their commitment, and the families for their willingness to participate in the study.

## Author contributions

**Conceptualization:** Renata Oliveira Neves, Leandro Meirelles Nunes, Juliana Rombaldi Bernardi.
**Data curation:** Renata Oliveira Neves, Cátia Regina Ficagna, Christy Hannah Sanini Belin, Larissa de Oliveira Silveira.
**Formal analysis:** Renata Oliveira Neves, Elma Izze da Silva Magalhães, Rogério Boff Borges, Juliana Rombaldi Bernardi.
**Funding acquisition:** Renata Oliveira Neves, Juliana Rombaldi Bernardi.
**Methodology:** Renata Oliveira Neves, Rogério Boff Borges, Leandro Meirelles Nunes, Juliana Rombaldi Bernardi.
**Project administration:** Renata Oliveira Neves, Juliana Rombaldi Bernardi.
**Supervision:** Leandro Meirelles Nunes, Juliana Rombaldi Bernardi.

**Validation:** Renata Oliveira Neves.

**Visualization:** Renata Oliveira Neves, Cátia Regina Ficagna, Paula Ruffoni Moreira, Christy Hannah Sanini Belin, Larissa de Oliveira Silveira.

**Writing – original draft:** Renata Oliveira Neves, Juliana Rombaldi Bernardi.

**Writing – review & editing:** Renata Oliveira Neves, Elma Izze da Silva Magalhães, Cátia Regina Ficagna, Paula Ruffoni Moreira, Christy Hannah Sanini Belin, Larissa de Oliveira Silveira, Rogério Boff Borges, Leandro Meirelles Nunes, Juliana Rombaldi Bernardi.

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
