## [Decision Letter · Decision Letter 0]

3 Jul 2025

PONE-D-25-08095Do children allocated to different methods of food introduction have distinct feeding preferences and flavor acceptance in the first years of life? A randomized clinical trialPLOS ONE?

Dear Dr. Bernardi,

Thank you for submitting your manuscript to PLOS ONE. After careful consideration, we feel that it has merit but does not fully meet PLOS ONE’s publication criteria as it currently stands. Therefore, we invite you to submit a revised version of the manuscript that addresses the points raised during the review process.

We look forward to receiving your revised manuscript.

Kind regards,

António Raposo

Academic Editor

PLOS ONE

Journal Requirements:

2. Thank you for submitting your clinical trial to PLOS ONE and for providing the name of the registry and the registration number. The information in the registry entry suggests that your trial was registered after patient recruitment began. PLOS ONE strongly encourages authors to register all trials before recruiting the first participant in a study.

1) your reasons for your delay in registering this study (after enrolment of participants started);

2) confirmation that all related trials are registered by stating: “The authors confirm that all ongoing and related trials for this drug/intervention are registered”.

“This research was supported by the Conselho Nacional de Desenvolvimento Científico e Tecnológico (CNPq), Brazil; Fundo de Incentivo à Pesquisa e Eventos (FIPE), Hospital de Clínicas de Porto Alegre, Brazil [grant number 2019-0540]; Coordenação de Aperfeiçoamento de Pessoal de Nível Superior (CAPES), Brazil [grant number 407426/2021-3].”

5. In the online submission form, you indicated that “Data will be made available on request.”. 

All PLOS journals now require all data underlying the findings described in their manuscript to be freely available to other researchers, either:

3. Uploaded as supplementary information.

7. We note that the original protocol that you have uploaded as a Supporting Information file contains an institutional logo. As this logo is likely copyrighted, we ask that you please remove it from this file and upload an updated version upon resubmission.

Reviewers' comments:

Reviewer's Responses to Questions

**Comments to the Author**

1. Is the manuscript technically sound, and do the data support the conclusions?

Reviewer #1: Partly

Reviewer #2: Partly

2. Has the statistical analysis been performed appropriately and rigorously?

Reviewer #1: Yes

Reviewer #2: No

3. Have the authors made all data underlying the findings in their manuscript fully available?

Reviewer #1: Yes

Reviewer #2: No

4. Is the manuscript presented in an intelligible fashion and written in standard English?

Reviewer #1: Yes

Reviewer #2: Yes

Reviewer #1: General Comments

This manuscript addresses a relevant and timely research question, examining how different methods of complementary food introduction—Parent-Led Weaning (PLW), Baby-Led Introduction to Solids (BLISS), and a Mixed approach—affect food preferences and taste acceptance among young children. Given the increasing interest from both clinicians and parents regarding early dietary influences on later eating behaviors, the study's objective is particularly valuable. The authors have employed a robust randomized clinical trial (RCT) design and clearly defined interventions, which significantly strengthens the methodological rigor.

The manuscript is generally well-structured, clearly written, and includes appropriate references to prior literature. Methodological details, including randomization, blinding, and statistical analyses (robust Poisson regression adjusted via Directed Acyclic Graph \[DAG] and intention-to-treat analyses), are suitably described. The combination of parent-reported feeding preferences and an objective laboratory taste acceptance test is commendable, adding depth and rigor to the findings.

However, several issues warrant attention to enhance clarity, accuracy, and appropriate interpretation. Below I outline key points that require revisions, categorized as major and minor issues.

Major Issues

1. Clarification of Primary and Secondary Outcomes

It is currently unclear which specific outcome measures were identified as primary or secondary prior to study initiation. The manuscript evaluates multiple outcomes across taste categories and groups, raising the potential for Type I error due to multiple comparisons. It would improve clarity to explicitly state the primary outcome(s) designated a priori, or clearly acknowledge that the analyses are exploratory. Additionally, authors should discuss potential risks of Type I errors explicitly, especially given that significant findings were limited to sour taste preferences.

2. Details and Transparency of Statistical Analysis

The authors employed robust Poisson regression appropriately, but clarity regarding adjusted versus unadjusted results is needed. The manuscript should explicitly state the covariates used in adjusted models and clearly indicate in the tables and results section which comparisons were adjusted or unadjusted. This transparency will enhance rigor and interpretability.

3. Handling of Missing Data and Attrition

Attrition, particularly in the Taste Acceptance Test (with approximately 34% non-completion), raises concerns about potential bias. The authors should detail reasons for attrition clearly within the participant flow diagram and address explicitly whether attrition was balanced across groups. Additionally, discuss whether attrition may have introduced bias, potentially influencing outcomes. A brief comparison of baseline characteristics between completers and non-completers would further strengthen methodological transparency.

4. Measurement Validity and Limitations

Additional detail on the structure, validity, and potential biases of the Feeding Preferences Questionnaire is necessary. The subjective nature of parent-reported preferences and the possibility of reporting bias should be explicitly addressed. Furthermore, the authors should clarify the limitations of the Taste Acceptance Test, emphasizing that laboratory responses may not fully capture everyday food acceptance behaviors.

5. Interpretation of Findings and Practical Significance

The authors' interpretation of the findings should be tempered. Given the limited number of significant differences (only sour taste preference showed significance), conclusions suggesting broad impacts of feeding methods should be revised to reflect the modest and limited scope of the observed effects. Additionally, the manuscript should discuss the potential practical implications of these small observed differences clearly and cautiously.

6. Sample Size and Power Considerations

Clarify explicitly the primary endpoint used for the original power calculation. If taste preference was not the primary outcome, clearly acknowledge the limitations this places on detecting subtle between-group differences in taste acceptance. This clarification will aid in interpreting the largely null findings for most taste categories.

7. Adherence to Assigned Feeding Methods

The manuscript should briefly summarize adherence to feeding methods as previously published by the authors, clearly discussing how actual adherence levels may have influenced study outcomes. Recognizing variability in adherence provides essential context for interpreting the subtle differences observed.

Minor Issues

1. Language and Terminology

* Replace "mensal" with "monthly" for clarity and correct terminology.

* Clarify clearly on first mention that the "Mixed" method involved initially using BLISS but switching to spoon-feeding if necessary.

2. Data Presentation and Table Corrections

* Correct the error in Table 2 (Mixed method salty taste PR value of "0.10" should likely be "1.00"). Verify consistency in decimal places and formatting.

3. Figure and Flowchart

* Provide a clear, detailed participant flowchart illustrating numbers randomized, completed, and lost for each group.

4. Discussion Contextualization

* Emphasize explicitly the transient and limited influence early feeding methods may have on taste preferences by referencing relevant longitudinal studies and aligning findings with existing literature.

5. References and Formatting

* Use consistent terminology for "Parent-Led Weaning" and "traditional method" to avoid confusion.

The manuscript presents valuable and rigorously conducted research with the potential for publication. Addressing the outlined major and minor issues will significantly improve clarity, methodological transparency, and appropriate interpretation, thus aligning the manuscript fully with high-quality standards required for acceptance.

Reviewer #2: Abstract: need to spell out the abbreviation of “PR” for the first time.

Sample size: what test was used? Which one is the primary outcome? The calculation is based on a continuous measure (standard deviation), however, all the outcomes in Table 2 are binary. It is not clear how the 1.4 SD was selected.

The p values need to be adjusted for multiple tests, especially for Table 2.

**Do you want your identity to be public for this peer review?** For information about this choice, including consent withdrawal, please see our Privacy Policy

Reviewer #1: No

Reviewer #2: No

---

## [Author Response · Author response to Decision Letter 1]

3 Sep 2025

Porto Alegre (RS), Brazil, September 01, 2025

Dear Academic Editor and Reviewers,

The authors would like to thank you for the careful evaluation of our manuscript entitled “Do children allocated to different methods of complementary feeding introduction have distinct food preferences and flavor acceptance in the first years of life? A randomized clinical trial” and for the comments and suggestions provided. We appreciate the opportunity to revise our manuscript and have addressed all the points raised in the review process.

Below, we provide a point-by-point response to the journal requirements and each comment from the reviewers and the academic editor. Our responses are presented in blue and indicate where the changes have been made in the revised manuscript (file: “Revised Manuscript with Track Changes”).

We believe that the changes made have significantly improved the manuscript and hope that our revised version meets the journal’s publication criteria and is now suitable for publication in PLOS ONE.

Finally, we would like to note that one additional author has been included in this version of the manuscript, given her substantial contribution to the construction of the statistical models required for the revision, as well as her participation in writing, editing and reviewing the manuscript. We would also like to respectfully inform that, for reasons of logistical convenience, the corresponding author is the undersigned of this letter.

Kind regards,

Cátia Regina Ficagna

JOURNAL REQUIREMENTS:

Response: We review the manuscript formatting according to the provided style templates, including file naming, to ensure it fully meets PLOS ONE style requirements.

2. Thank you for submitting your clinical trial to PLOS ONE and for providing the name of the registry and the registration number. The information in the registry entry suggests that your trial was registered after patient recruitment began. PLOS ONE strongly encourages authors to register all trials before recruiting the first participant in a study.

1) your reasons for your delay in registering this study (after enrolment of participants started);

2) confirmation that all related trials are registered by stating: “The authors confirm that all ongoing and related trials for this drug/intervention are registered”.

Response: After study approval by the Research Ethics Committee, the RCT was registered at https://ensaiosclinicos.gov.br/rg/RBR-229scm, identifier [RBR-229scm U1111-1226-9516]. Due to administrative and logistical challenges during the initial phase of the study’s preparation, there was a delay in registering this clinical trial.. Specifically, the registration process took longer than anticipated due to the time required to obtain internal approvals and complete the study documentation. This explanation has now been included under “Registration” in the Methods section. In addition, we confirm that all related trials have been registered, including the requested declaration, as follows:

“Registration

After study approval by the Research Ethics Committee, the RCT was registered at https://ensaiosclinicos.gov.br/rg/RBR-229scm, identifier [RBR-229scm U1111-1226-9516]. The authors confirm that all ongoing and related trials for this intervention are registered.”

Response: Thank you for pointing this out. Indeed, the grant number from CNPq (grant number: 20190230) was missing in the Financial Disclosure section. We have now added the correct grant number during the resubmission and ensured that the information matches in both the “Funding Information” and “Financial Disclosure” sections.

“This research was supported by the Conselho Nacional de Desenvolvimento Científico e Tecnológico (CNPq), Brazil; Fundo de Incentivo à Pesquisa e Eventos (FIPE), Hospital de Clínicas de Porto Alegre, Brazil [grant number 2019-0540]; Coordenação de Aperfeiçoamento de Pessoal de Nível Superior (CAPES), Brazil [grant number 407426/2021-3].”

Response: We thank you for your comment. We confirmed that the funders had no role in study design, data collection and analysis, decision to publish, or preparation of the manuscript, and this will be confirmed at the time of submission.

5. In the online submission form, you indicated that “Data will be made available on request.”. 

All PLOS journals now require all data underlying the findings described in their manuscript to be freely available to other researchers, either:

3. Uploaded as supplementary information.

Response: We thank you for your comment. The dataset generated and analyzed during the current study will be made publicly available as supplementary information upon resubmission, in accordance with PLOS ONE data availability requirements. No ethical or legal restrictions prevent the sharing of this dataset.

Response: We thank you for your comment. We would include full captions for all Supporting Information files at the end of the manuscript and update the in-text citations accordingly, in accordance with the PLOS ONE Supporting Information guidelines.

7. We note that the original protocol that you have uploaded as a Supporting Information file contains an institutional logo. As this logo is likely copyrighted, we ask that you please remove it from this file and upload an updated version upon resubmission.

Response: We thank you for your comment. We would remove the institutional logo from the original protocol file and upload an updated version as Supporting Information upon resubmission.

REVIEWERS' COMMENTS

REVIEWER #1:

General Comments: This manuscript addresses a relevant and timely research question, examining how different methods of complementary food introduction—Parent-Led Weaning (PLW), Baby-Led Introduction to Solids (BLISS), and a Mixed approach—affect food preferences and taste acceptance among young children. Given the increasing interest from both clinicians and parents regarding early dietary influences on later eating behaviors, the study's objective is particularly valuable. The authors have employed a robust randomized clinical trial (RCT) design and clearly defined interventions, which significantly strengthen the methodological rigor. The manuscript is generally well-structured, clearly written, and includes appropriate references to prior literature. Methodological details, including randomization, blinding, and statistical analyses (robust Poisson regression adjusted via Directed Acyclic Graph \[DAG] and intention-to-treat analyses), are suitably described. The combination of parent-reported feeding preferences and an objective laboratory taste acceptance test is commendable, adding depth and rigor to the findings.

However, several issues warrant attention to enhance clarity, accuracy, and appropriate interpretation. Below, I outline key points that require revisions, categorized as major and minor issues.

Major Issues:

1) Clarification of Primary and Secondary Outcomes: It is currently unclear which specific outcome measures were identified as primary or secondary prior to study initiation. The manuscript evaluates multiple outcomes across taste categories and groups, raising the potential for Type I error due to multiple comparisons. It would improve clarity to explicitly state the primary outcome(s) designated a priori or clearly acknowledge that the analyses are exploratory. Additionally, authors should discuss potential risks of Type I errors explicitly, especially given that significant findings were limited to sour taste preferences.

Response: We appreciate the comment. As described in the randomized clinical trial protocol (Nunes et al., 2021), the primary outcomes were z-scores of anthropometric measurements (weight for age, weight for length, length for age, and BMI for age z-scores) to investigate child growth and nutritional status. Secondary outcomes included food and flavor preferences, among other outcomes (such as choking prevalence, dietary variety, child and parent feeding behavior, iron deficiency, oral hygiene behavior, dental caries, dental development, gingival health status, the prevalence of functional constipation, maternal perception of the methods of CF introduction, and prevalence of child eating disorders). To clarify this, we revised the Methods section of the manuscript and included an “Outcomes” subsection to explicitly indicate that food and flavor preferences were assessed as a priori-defined secondary outcomes, as follows:

“Outcomes

The RCT had as primary outcomes the z-scores of anthropometric measurements (weight for age, weight for length, length for age, and BMI for age z-scores) [9,10] to investigate child growth and nutritional status [18]. The secondary outcomes, defined a priori in the RCT, included, among others (prevalence of choking, dietary variety, child and parental eating behavior, iron deficiency, oral hygiene behavior, dental caries, dental development, gingival health, prevalence of functional constipation, maternal perception of the methods of CF introduction, and prevalence of childhood eating disorders), food and flavor preferences [9,10], which were investigated as outcomes in the present study.”

18. Ficagna CR, Magalhães EIDS, Chacón AB, Moreira PR, Neves RO, Nunes LM, Bernardi JR. Impact of complementary feeding methods on infant nutritional status: A randomized clinical trial. Nutrition. 2025 Jul 11;140:112908. doi: 10.1016/j.nut.2025.112908. Epub ahead of print. PMID: 40782569.

Regarding the potential risks of Type I errors, we included in the Discussion section a note about the risk of these errors due to multiple comparisons, highlighting that the significant findings were limited to the sour taste, as below:

“On the other hand, given the multiple comparisons performed, there is an increased risk of Type I error. This is particularly relevant as significant findings were limited to sour taste preferences, which should be interpreted with caution.”

2) Details and Transparency of Statistical Analysis: The authors employed robust Poisson regression appropriately, but clarity regarding adjusted versus unadjusted results is needed. The manuscript should explicitly state the covariates used in adjusted models and clearly indicate in the tables and results section which comparisons were adjusted or unadjusted. This transparency will enhance rigor and interpretability.

Response: We appreciate the comment. As requested, we revised the text of the Statistical Analyses in the Methods section to clarify the covariates used in the adjusted models. We also revised the text of the Results section, as well as Tables 2 and 4, to clearly indicate the adjusted and unadjusted results. Below is the revised text of the Methods and Results sections:

Methods section:

“Statistical Analysis

All analyses were performed by intention-to-treat. The database was created utilizing the SPSS® Program – Statistical Package for the Social Sciences version 29.0, with double entry and subsequent validation.

Categorical variables were described using absolute and relative frequencies [(n (%)] and numeric variables were described as measures of central tendency and dispersion (mean and standard deviation; median and P25 - P75). The distribution of numeric variables was assessed through visual inspection of histograms, verification of asymmetry and kurtosis measures, as well as the application of the Shapiro-Wilk test. To compare the sociodemographic, nutritional, and childcare-related characteristics of the study sample between the intervention groups (PLW, BLISS and Mixed), Pearson's chi-square test was used for the proportions of categorical variables, and the Mann-Whitney test was used for the medians of numerical variables. The Mann-Whitney test was also used to compare the medians of the volume consumed in the taste acceptance test according to the infant's reaction (Acceptance and rejection) to the same substances. A significance level of 5% (p ≤ 0.05) was considered for all analyses.

Poisson regression analysis with robust variance was used to estimate prevalence ratios and their 95% confidence intervals for the associations between the method of CF introduction and food preferences according to the predominant taste, as well as between food preferences according to the predominant taste and reactions in the taste acceptance test. To control for confounding, Poisson regression analysis was adjusted for potential confounding factors. A minimum adjustment set of variables to control for confounding in each association evaluated was identified based on a directed acyclic graph (DAG) constructed using the online DAGitty 3.2 software, applying the backdoor criterion [24]. The DAG for estimating the total effect of the method of CF introduction on food preferences according to the predominant taste indicated the need for adjustment for: Maternal education, parity, maternal BMI, daycare attendance and child’s primary caregiver (Supplementary Figure 1) . The DAG for estimating the total effect of the food preferences according to the predominant taste on reactions in the taste acceptance test, in turn, indicated the need for adjustment for: Maternal education, maternal BMI, age at CF introduction, method of CF introduction, age at the application of the food preferences questionnaire and taste acceptance, daycare attendance, child’s primary caregiver and type of milk consumed (Supplementary Figure 2).”

Results section:

Lines 305 to 314:

“The crude and adjusted prevalence ratios of the association between the method of CF introduction and the food preferences according to the predominant taste are shown in Table 2. In adjusted analysis, the prevalence of preferences for foods with a predominant sour taste were 23% higher (Crude PR 1.23, 95% CI 1.03 - 1.47, p=0.020) in the Mixed method compared to the PLW, however, after adjusting for covariates, this association did not remain statistically significant (Adjusted PR 1.15, 95% CI 0.94 - 1.41, p=0.173). The preference for foods with other predominant tastes didn’t show associations with the method of CF introduction, neither in the unadjusted nor in the adjusted analysis.”

Lines 331 to 337:

“The crude and adjusted PR of the association between the f

---

## [Decision Letter · Decision Letter 1]

14 Sep 2025

PONE-D-25-08095R1Do children allocated to different methods of complementary feeding introduction have distinct food preferences and flavor acceptance in the first years of life? A randomized clinical trialPLOS ONE?

Dear Dr. Bernardi,

Thank you for submitting your manuscript to PLOS ONE. After careful consideration, we feel that it has merit but does not fully meet PLOS ONE’s publication criteria as it currently stands. Therefore, we invite you to submit a revised version of the manuscript that addresses the points raised during the review process.

We look forward to receiving your revised manuscript.

Kind regards,

António Raposo

Academic Editor

PLOS ONE

Journal Requirements:

Reviewers' comments:

Reviewer's Responses to Questions

**Comments to the Author**

Reviewer #1: All comments have been addressed

Reviewer #2: All comments have been addressed

2. Is the manuscript technically sound, and do the data support the conclusions?

Reviewer #1: Yes

Reviewer #2: (No Response)

3. Has the statistical analysis been performed appropriately and rigorously?

Reviewer #1: Yes

Reviewer #2: (No Response)

4. Have the authors made all data underlying the findings in their manuscript fully available?

Reviewer #1: Yes

Reviewer #2: (No Response)

5. Is the manuscript presented in an intelligible fashion and written in standard English?

Reviewer #1: Yes

Reviewer #2: (No Response)

Reviewer #1: General Comments

The revised manuscript represents a clear improvement over the initial submission. The authors have carefully and systematically addressed the major and minor points raised during the first review. The structure and readability are improved, the methods and results are clarified, and the conclusions are now more cautious and aligned with the data. The inclusion of additional details on outcomes, statistical methods, attrition, and adherence strengthens the manuscript’s transparency and rigor. The updated flowchart, supplementary materials, and data availability statements bring the submission into better compliance with PLOS ONE and CONSORT requirements.

The study remains a valuable contribution to the literature on complementary feeding methods and children’s later taste preferences. The novelty of including a “Mixed” approach, the dual use of subjective (questionnaire) and objective (taste acceptance test) measures, and the randomized trial design make the research relevant to pediatric nutrition and public health.

Major Issues – Status of Revisions

1.Clarification of Primary vs. Secondary Outcomes

Addressed. The authors now explicitly state that the trial’s primary outcomes were anthropometric z-scores, while taste preferences were pre-specified secondary outcomes. The Discussion now acknowledges Type I error risk from multiple comparisons.

2.Statistical Transparency (Adjusted vs. Unadjusted Results)

Addressed. Tables 2 and 4 now present both crude and adjusted prevalence ratios, with covariates defined via DAGs. Methods and Results have been revised for clarity.

3.Missing Data and Attrition

Addressed. The flowchart and Supplementary Table 1 now describe reasons for attrition and compare completers vs. non-completers. The Discussion explicitly states that attrition bias is unlikely.

4.Validity and Limitations of Measurement Tools

Addressed. The Methods now better describe the Food Preferences Questionnaire and the Taste Acceptance Test. The Discussion highlights potential biases and limitations.

5.Interpretation and Practical Significance

Addressed. Conclusions are now tempered: only sour taste differences in crude analysis are acknowledged, and these did not hold after adjustment.

6.Sample Size and Power

Addressed. The Methods clarify that the sample size was powered for growth outcomes, not taste preferences, which were secondary outcomes.

7.Adherence to Feeding Methods

Addressed. The Discussion now incorporates prior adherence results (Sanini Belin et al., 2023) and acknowledges crossover between groups, which may explain null findings.

Minor Issues – Status of Revisions

#Terminology (“mensal” → “monthly”): Corrected.

#Clarification of “Mixed” method: Expanded.

#Table 2 error (“0.10”): Corrected to 0.99, decimals standardized.

#Flowchart: Revised and detailed.

#Contextualization in Discussion: Added references to longitudinal evidence, reinforcing transient effects.

#Terminology consistency (PLW vs. traditional): Corrected throughout.

#Abbreviation (“PR”): Defined on first use in abstract.

All minor issues appear to have been addressed satisfactorily.

Compliance with PLOS ONE Requirements

#Trial Registration: Reasons for delayed registration added; confirmation that related trials are registered.

#Funding Statements: Now consistent; funders’ lack of role explicitly stated.

#Data Availability: Full dataset uploaded as Supplementary Information.

#Supporting Information Captions: Added.

#Copyrighted Logos: Removed from protocol file.

The manuscript now meets PLOS ONE’s editorial and ethical standards.

Minor Editorial Adjustments

・Ensure consistent tense usage throughout the manuscript, particularly when describing methods (past tense) and results (past tense).

・Double-check supplementary figure captions to ensure they are placed correctly and consistently referenced in the main text.

・Standardize abbreviation usage (e.g., consistently use “CF” for complementary feeding after first definition).

・Review for minor typographical errors (spacing, punctuation, en-dash vs. hyphen in numerical ranges).

・Verify reference formatting consistency (especially newer references added in revision; ensure full journal names or abbreviations follow PLOS ONE style).

・Confirm that tables and figures have consistent decimal places, units, and legends across the manuscript and supplementary materials.

・Check the Acknowledgments section for grammatical refinement (e.g., “thank for the commitment” → “thank the researchers for their commitment”).

Reviewer #2: All concerns are addressed.

**Do you want your identity to be public for this peer review?** For information about this choice, including consent withdrawal, please see our Privacy Policy

Reviewer #1: No

Reviewer #2: No

---

## [Author Response · Author response to Decision Letter 2]

24 Sep 2025

Porto Alegre (RS), Brazil, September 24, 2025

Dear Academic Editor and Reviewers,

The authors would like to thank you for the careful evaluation of our manuscript entitled “Do children allocated to different methods of complementary feeding introduction have distinct food preferences and flavor acceptance in the first years of life? A randomized clinical trial” and for the comments and suggestions provided. We are encouraged that the reviewers considered our previous responses satisfactory.

In this revised version, we have implemented the Minor Editorial Adjustments requested. Below, we provide a point-by-point response to the journal requirements and each comment from the academic editor. Our responses are presented in blue and indicate where the changes have been made in the revised manuscript (file: “Revised Manuscript with Track Changes”).

The authors believe that the changes made have significantly improved the manuscript and hope that our revised version fully meets PLOS ONE’s publication criteria and is now suitable for publication.

Kind regards,

Cátia Regina Ficagna

Minor Editorial Adjustments

1. Ensure consistent tense usage throughout the manuscript, particularly when describing methods (past tense) and results (past tense).

Response: As requested, we reviewed the entire manuscript and adjusted the text to ensure consistency of verb tense throughout, using the past perfect tense, particularly in the Methods and Results sections.

2. Double-check supplementary figure captions to ensure they are placed correctly and consistently referenced in the main text.

Response: We checked the supplementary figure captions and inserted them in the text of the manuscript, following the paragraph in which the figure is first cited.

3. Standardize abbreviation usage (e.g., consistently use “CF” for complementary feeding after the first definition).

Response: We revised the manuscript text and standardized the abbreviation usage throughout, consistently using “CF” for complementary feeding after its first definition.

4. Review for minor typographical errors (spacing, punctuation, en-dash vs. hyphen in numerical ranges).

Response: We have reviewed the manuscript for minor typographical errors, including spacing, punctuation, and the correct use of en-dash versus hyphen in numerical ranges, and made necessary corrections.

5. Verify reference formatting consistency (especially newer references added in revision; ensure full journal names or abbreviations follow PLOS ONE style).

Response: We have reviewed the reference list, ensuring consistency in formatting. All references, including those newly added in this revision, now follow PLOS ONE style, with full journal names or abbreviations correctly applied.

6. Confirm that tables and figures have consistent decimal places, units, and legends across the manuscript and supplementary materials.

Response: We have reviewed all tables and figures in the manuscript and supplementary materials and made the necessary corrections to ensure consistent decimal places, units, and legends throughout.

7. Check the Acknowledgments section for grammatical refinement (e.g., “thank for the commitment” → “thank the researchers for their commitment”).

Response: We have revised the Acknowledgments section for grammatical refinement, as below:

“The authors thank the researchers for their commitment, and the families for their willingness to participate in the study.”

---

## [Decision Letter · Decision Letter 2]

15 Oct 2025

Do children allocated to different methods of complementary feeding introduction have distinct food preferences and flavor acceptance in the first years of life? A randomized clinical trial

PONE-D-25-08095R2

Dear Dr. Bernardi,

We’re pleased to inform you that your manuscript has been judged scientifically suitable for publication and will be formally accepted for publication once it meets all outstanding technical requirements.

Kind regards,

António Raposo

Academic Editor

PLOS ONE

Additional Editor Comments (optional):

Reviewers' comments:

Reviewer's Responses to Questions

**Comments to the Author**

Reviewer #1: All comments have been addressed

Reviewer #2: All comments have been addressed

2. Is the manuscript technically sound, and do the data support the conclusions?

Reviewer #1: Yes

Reviewer #2: (No Response)

3. Has the statistical analysis been performed appropriately and rigorously?

Reviewer #1: Yes

Reviewer #2: (No Response)

4. Have the authors made all data underlying the findings in their manuscript fully available?

Reviewer #1: Yes

Reviewer #2: (No Response)

5. Is the manuscript presented in an intelligible fashion and written in standard English?

Reviewer #1: Yes

Reviewer #2: (No Response)

Reviewer #1: I would like to sincerely commend the authors for their thoughtful and thorough revisions throughout this review process. Each of the points raised in the previous rounds has been carefully and transparently addressed. The manuscript now reads smoothly, with improved clarity, consistent terminology, and precise alignment between the data and conclusions.

The study remains an important and well-executed randomized clinical trial exploring an underexamined but meaningful question in pediatric nutrition—the influence of different complementary feeding methods on early taste preferences. The methodological rigor and balanced interpretation make this paper a valuable contribution to the field.

Thank you for your sustained efforts and patience over multiple rounds of revision. Your diligence has clearly strengthened the quality and readability of the final manuscript. Congratulations on your excellent work—I am pleased to recommend it for publication.

Reviewer #2: All concerns are addressed.

**Do you want your identity to be public for this peer review?** For information about this choice, including consent withdrawal, please see our Privacy Policy

Reviewer #1: No

Reviewer #2: No

---

## [Editor Report · Acceptance letter]

PONE-D-25-08095R2

PLOS ONE

Dear Dr. Bernardi,

I'm pleased to inform you that your manuscript has been deemed suitable for publication in PLOS ONE. Congratulations! Your manuscript is now being handed over to our production team.

Kind regards,

on behalf of

Dr. António Raposo

Academic Editor

PLOS ONE